# The Relationship between the Timing of Sugammadex Administration and the Upper Airway Obstruction during Awakening from Anesthesia: A Retrospective Study

**DOI:** 10.3390/medicina57020088

**Published:** 2021-01-21

**Authors:** Eunsu Kang, Byeong Cheol Lee, Jae Hong Park, Sang Eun Lee, Se Hun Kim, Daeseok Oh, Dae Yun Choi, Myoung Jin Ko

**Affiliations:** Department of Anesthesiology and Pain Medicine, Haeundae Paik Hospital of Inje University, 875 Haeun-daero, Haeundae-gu, Busan KS012, Korea; iceneco1120@gmail.com (E.K.); ironism00@gmail.com (B.C.L.); H00150@paik.ac.kr (J.H.P.); painlse@gmail.com (S.E.L.); anesehunkim@outlook.kr (S.H.K.); yivangin@naver.com (D.O.); H80639@paik.ac.kr (D.Y.C.)

**Keywords:** sugammadex, minimum alveolar concentration, upper airway obstruction

## Abstract

*Background and Objectives*: The harmonization of recovery of consciousness and muscular function is important in emergence from anesthesia. Even if muscular function is recovered, tracheal extubation without adequate recovery of consciousness may increase the risk of respiratory complications. In particular, upper airway obstruction is one of the common respiratory complications and can sometimes be fatal. However, the association between the timing of sugammadex administration and the upper airway obstruction that can occur during awakening from anesthesia has rarely been studied. *Materials and Methods*: The medical records of 456 patients who had surgery under general endotracheal anesthesia (GETA) at the Haeundae Paik Hospital between October 2017 and July 2018 and who received intravenous sugammadex to reverse rocuronium-induced neuromuscular blockade were analyzed. The correlations between bispectral index (BIS) and minimum alveolar concentration (MAC) at the time of sugammadex administration, the incidence of complications, and the time to tracheal extubation were analyzed to investigate how different timings of sugammadex administration affected upper airway obstruction after tracheal extubation. *Conclusions*: The effect of BIS and the duration from anesthetic discontinuation to sugammadex administration on upper airway obstruction was not statistically significant. However, the odds ratio of complication rates with MAC < 0.3 compared with MAC ≥ 0.3 was 0.40 (95% confidence interval 0.20 to 0.81, *p* = 0.011), showing a statistically significant increase in risk with MAC ≥ 0.3 for upper airway obstruction.

## 1. Introduction

The onset of action of any medication and its interaction with other drugs should be considered to maximize its therapeutic effects and minimize adverse effects [1]. As such, there is an optimal time for the administration of any medication, particularly those used for reversing the effects of muscle relaxants after general endotracheal anesthesia (GETA) since they are known to have optimal action times.

Sugammadex is a newly developed drug that reverses muscle relaxants. This is a modified γ-cyclodextrin and a selective relaxant–binding agent used to inactivate specific nondepolarizing neuromuscular blocking agents, such as a rocuronium. The required doses according to the degree of muscle relaxation of sugammadex are widely known, but the relationship between the residual inhalation anesthetic or the sedation level and the timing of sugammadex administration is not well known. In most patients, all different levels of neuromuscular blockade are reversed within 2 to 3 min after administration of sugammadex [2,3]. This rapid reversal of neuromuscular blockade using sugammadex may affect emergence depending on the residual inhalation anesthetics. Airway obstruction, which can occur after extubation, is a common complication but can sometimes be fatal [4]. However, the effects of administering sugammadex at different anesthetic depths on upper airway obstruction after extubation have not been studied. Therefore, we conducted this study to find out the relationship between the timing of administration of sugammadex and the occurrence of upper airway obstruction.

## 2. Materials and Methods

This retrospective study was conducted at the Haeundae Paik Hospital of Inje University, and it was approved by the Institutional Review Board (IRB) and registered at https://cris.nih.go.kr (protocol number KCT0003356) and the requirement for written informed consent was waived by the IRB. All patients who had surgery under GETA using inhalation anesthetics at the Haeundae Paik Hospital between October 2017 and July 2018 and who received intravenous sugammadex to reverse rocuronium-induced neuromuscular blockade were included in this study. We reviewed electronic medical records, including preoperative evaluations, anesthesia records, and postoperative care unit (PACU) records and nursing information surveys, in order to collect data on bispectral index (BIS) (BisTMQuatro, Covidien Inc., Singapore) and minimum alveolar concentration (MAC) (Primus^®^, Dräger Medical, Lübeck, Germany) of volatile anesthetics at the time of sugammadex administration. The age, sex, height, weight, American Society of Anesthesiology Physical Status (ASA PS) score, diagnosis, type of surgery, underlying diseases, medical and medication history, type of inhalation anesthetics (sevoflurane or desflurane), total dose of rocuronium administered, total dose of sugammadex, operation duration, anesthesia duration, time when an inhalation anesthetic was discontinued (T1), time of sugammadex administration (T2), and time of tracheal extubation (T3) were also recorded. The dose of sugammadex was determined by the anesthesiologist taking into consideration the time of the last administration of rocuronium, the total amount of rocuronium, and the patient’s response. Using these data, we investigated the distribution in durations between anesthetic discontinuation and sugammadex administration (T2-T1, min) as well as whether these durations have any correlation with the incidence of respiratory complications after extubation. To investigate how the different timings of sugammadex administration affected the respiratory complications, we analyzed the correlations between BIS and MAC at the time of sugammadex administration, the incidence of complications, and the time to tracheal extubation (T3-T1, min). The primary outcome for our study was the incidence of upper airway obstruction according to the timing of administration of sugammadex. We defined upper airway obstruction as the necessity for an airway device or manual intervention to maintain the airway listed in the medical record, such as continuous positive pressure or manual positive pressure ventilation.

The patients whose parameters were missing in the electronic medical records, those who had surgery under total intravenous anesthesia, those who received other reversal agents (e.g., neostigmine or pyridostigmine), and those who were administered sugammadex after tracheal extubation were excluded from this study.

### Statistical Analysis

We presented the data as frequency and percentage for categorical variables and mean ± standard deviation (SD) for numeric variables. The differences in study participants’ characteristics were compared across subgroups using the Chi-square test or Fisher’s exact test for categorical variables and using the independent test or the Mann–Whitney U test for continuous variables as appropriate. To check normal distribution, we used the Shapiro–Wilk test.

The effect of independent variables on response variables was analyzed using the univariate and multivariate linear regression, and the statistically significant variables were selected by the backward elimination method using a 0.05 alpha level. To check the multicollinearity problem, the variance inflation factor (VIF) was also estimated. The VIF quantifies the severity of multicollinearity in regression analysis and a VIF < 10 indicated that there was no problematic multicollinearity among the independent variables (VIF < 10). We set complication as a dependent variable. The corresponding odds ratio of each independent variable with a 95% confidence interval and an equivalent *p*-value were calculated with a multivariate logistic regression analysis method. Variables were selected using the backward elimination method using a 0.05 alpha level.

All statistical analyses were carried out using the SPSS 24.0 software and *p* values < 0.05 were considered significant.

## 3. Results

A total of 3391 patients were administered sugammadex during the period. Based on our inclusion criteria, we included a total of 456 patients in our study. The patients whose parameters were missing in the electronic medical records (*n* = 2712), those who had surgery under total intravenous anesthesia (*n* = 177), those who received other reversal agents concomitantly (*n* = 32), and those who were administered sugammadex after tracheal extubation (*n* = 14) were excluded from this study. The patients’ basic characteristics are shown in Table 1. The distribution in durations between anesthetic discontinuation and sugammadex administration (T2-T1, min) were also investigated, and the median value was 2 min (Figure 1). The patients were classified into two groups for each independent variable using its median value: sugammadex administration 2 min before or after discontinuation of inhalation anesthetics, BIS value less than or greater than 60, and MAC value less than or greater than 0.3 MAC. The results were analyzed and compared between the groups. Other factors, including sex, age, myocardial infarction (MI), underlying diseases, type of inhalation anesthetics, duration of surgery, duration of anesthesia, American Society of Anesthesiology (ASA) physical status, emergency surgery, and the total amount of rocuronium used, were also used as independent variables for statistical analyses. The events of upper airway obstruction were recorded in 39 patients (8.55%). We regarded these records as the events of upper airway obstruction (Table 2). The effect of T2-T1 on the complication rate was not statistically significant (univariate regression analysis, *p* = 0.03). Also, the total dose of rocuronium or sugammadex did not affect the occurrence of complications. However, the odds ratio of complication rates with MAC < 0.3 compared with MAC ≥ 0.3 was 0.40 (95% confidence interval 0.20 to 0.81, *p* = 0.011), showing a statistically significant increased risk with MAC ≥ 0.3 for complications. All patients with complications recovered immediately after appropriate treatment in the early stage, and there were no other serious sequelae, and there were no patient complaints of recall of memory during awakening from anesthesia.

T3-T1 was significantly reduced in the patients younger than 60 (0.76 min shorter than that of patients aged ≥60, *p* < 0.05). T3–T1 significantly increased if sevoflurane was used for anesthesia (2.39 min longer than desflurane, *p* < 0.05) or the duration of surgery was prolonged (*p* < 0.05) (Table 3).

## 4. Discussion

We investigated the association between the timing of sugammadex administration and the emergence from anesthesia and found that upper airway obstruction after reversal of muscle paralysis was reduced with sugammadex administration at an MAC value <0.3. Recovery of neuromuscular function and consciousness are two important factors in the recovery from GETA, and their harmony is essential for a smooth awakening. If only the neuromuscular functions have been restored, a patient may not cooperate, leading to a high probability of respiratory complications [5,6]. Conversely, a patient may experience severe anxiety and fear with the recovery of consciousness alone. This study aimed to identify the appropriate timing of the administration of sugammadex in relation to the depth of the anesthesia, which may reduce complications during emergence from anesthesia, decrease the duration of emergence, and stabilize anxiety. Data on time after discontinuation of anesthesia, BIS values, and MAC values were collected from the medical records. A median value of the T2-T1 distribution was shown to be approximately 2 min. We analyzed the effects of T2-T1 on the outcomes using two distinct intervals, duration of T2-T1 < 2 min and T2-T1 > 2 min, but no effect was observed. The T3-T1 interval increased with increasing T2-T1 interval, but these findings were not included in the results since the T3-T1 interval included the T2-T1 interval. In the previous prospective study in which sugammadex was administered in the early phase and the late phase (i.e., immediately after surgery vs. 2 min after the discontinuation of sevoflurane), the frequency of laryngeal spasm was shown to be significantly high in the early group [7]. In our study, however, no difference in complication occurrence was shown between the two groups. This inconsistency is probably due to the differences in the study designs; this was a retrospective study, while the previous study was a prospective controlled study. Some conditions, such as the concentration of inhalation anesthetics before discontinuation or the fresh-gas flow rates, which may affect the time to wash out after discontinuation, were not controlled. The anesthetic concentration that remained at the time of sugammadex administration could be significantly different even with the same duration of T2-T1 [8].

A BIS monitor analyzes patterns of electroencephalogram to assess the level of sedation. The BIS value is maintained between 40 and 60 under GETA, and the value > 60 may indicate a risk for anesthesia awareness [9]. We set the upper limit of the depth of anesthesia as 60 when defining our study groups. The BIS values at the time of sugammadex administration were shown to have no effect on complications, and emergence times. However, our results need further validation because the BIS values between 60 and 80 represent a state of light sedation in which the risks for complications during extubation become high. Previous studies have reported that extubation could be performed more safely on fully awake patients than on those who were lightly sedated [6]. It is unlikely that administering sugammadex at BIS values between 60 and 80 would lower the risks for extubation complications, but it might be meaningful to analyze the effect of BIS values > 80 on these risks, considering that the muscular blockade is reversed within 2 min after the administration. Our study unfortunately had only a few cases for this analysis. Additionally, the BIS value may not accurately reflect the immediate level of sedation during sugammadex administration since it is influenced by muscle tone and has a time delay [10,11].

In 1965, Eger et al. introduced the concept of MAC, relating the concentration or partial pressure of volatile anesthetics to a single clinically relevant end point of GETA [12]. The definition of 1 MAC is the minimum alveolar concentration of a volatile anesthetic required to prevent movement in 50% of patients in response to a surgical incision. Stoelting et al. introduced the concept of MAC-awake, which was defined as the anesthetic concentration needed to suppress a voluntary response to verbal command in 50% of patients [13]. The MAC-awake of sevoflurane and desflurane are approximately 0.3 MAC [13,14]. The MAC values displayed on the monitors of the Primus^®^ anesthesia machine were calculated based on the end-tidal inhalation anesthetic concentration and each patient’s age using the formula designed by Mapleson [15]. The MAC value, which enables the estimation of the depth of anesthesia, is based on the concentration of the inhalation anesthetic, and it is frequently used in the maintenance of GETA [13,14,16,17]. We found that the incidence of upper airway obstruction was significantly decreased when an end-tidal inhalation anesthetic gas concentration < 0.3 MAC with sugammadex administration was used compared with that at end-tidal gas concentrations > 0.3 MAC. However, the MAC values had no significant effect on the emergence time, but rather, the types of anesthesia, the age of the patient, and the operation time were found to have more influence on emergence. The time for complete recovery of consciousness was longer with sevoflurane than with desflurane, with increasing patient age and operation duration. These findings were consistent with the results of previous studies [18,19,20]. Regarding BIS values and T2-T1 intervals, no significant difference was found in the incidence of complications or other factors. Currently there is no direct antagonist for inhalation anesthetics; a full return to consciousness depends on slow drug elimination via respiration. Further studies are needed to harmonize the timing of recovery of consciousness and motor function.

This study had some limitations. Because it was a retrospective study, we could not control several variables that affect the outcome, such as the concentration of inhalation anesthetics for maintenance, types of surgery, and the amounts of sugammadex and rocuronium administered. In addition, association with the Train of four (TOF) ratio could not be analyzed since TOF monitoring was not routinely performed in our hospital.

## 5. Conclusions

In conclusion, we found that the incidence of upper airway obstruction after successful reversal of muscle paralysis could significantly be reduced when reversal with sugammadex and then extubation were performed with <0.3 MAC.

## Figures and Tables

**Figure 1 medicina-57-00088-f001:**
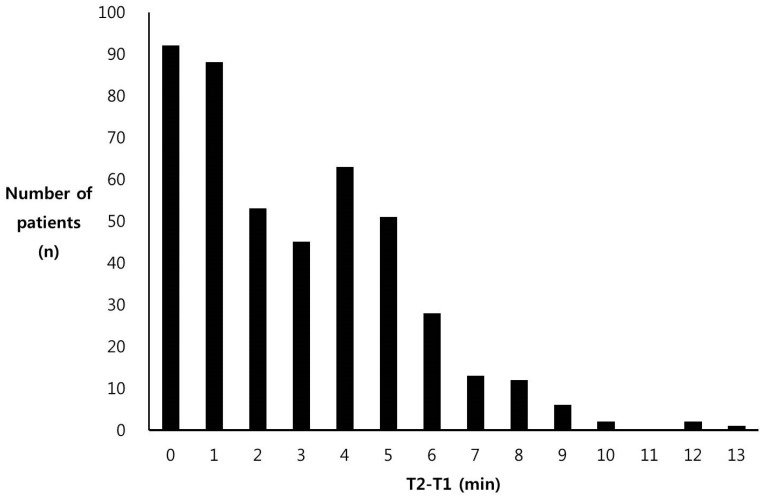
Distribution of patients according to the duration between anesthetic discontinuation and sugammadex administration (T2-T1, min). T2-T1 indicates the durations between anesthetic discontinuation and sugammadex administration.

**Table 1 medicina-57-00088-t001:** Baseline characteristics of patients.

Variable	Overall (*n* = 456)
**Sex**	
Male	213 (46.7)
Female	243 (53.3)
**Age**	
Mean ± SD	55.13 ± 15.17
<60	268 (58.8)
≥60	188 (41.2)
**DM**	
Yes	64 (14.0)
No	392 (86.0)
**Hypertension**	
Yes	112 (24.6)
No	344 (75.4)
**COPD**	
Yes	11 (2.4)
No	445 (97.6)
**Inhalational anesthetic**	
Desflurane	339 (74.3)
Sevoflurane	117 (25.7)
**Operation duration (min)**	
Mean ± SD	100.00 ± 80.39
**Anesthesia duration (min)**	
Mean ± SD	140.56 ± 89.07
**ASA PS**	
1,2	405 (88.8)
≥3	51 (11.2)
**Emergency surgery**	
Yes	31 (6.8)
No	425 (93.2)
**Total amount of rocuronium (mg)**	
Mean ± SD	60.92 ± 20.05
**BIS**	
<60	293 (64.3)
≥60	163 (35.7)
**MAC**	
<0.3	232 (50.9)
≥0.3	224 (49.1)

Abbreviations: SD, standard deviation; DM, diabetes mellitus; COPD, chronic obstructive lung disease; ASA PS, American Society of Anesthesiology physical status; BIS, bispectral index; MAC, minimum alveolar concentration. Values are presented as numbers (percent) or mean ± standard deviation.

**Table 2 medicina-57-00088-t002:** The events of upper airway obstruction after extubation.

	Events			Uni-Factor Analysis	
Variable	yes (*n* = 39)	no (*n* = 417)	*p*	OR (95% CI)	*p*
**T2-T1**					
≤2 min	25 (64.1)	208 (49.9)	0.089 ^1^	1.79 (0.91–3.55)	0.093
>2 min	14 (35.9)	209 (50.1)		ref	
**BIS**					
<60	28 (71.8)	265 (63.5)	0.304 ^1^	1.46 (0.71–3.02)	0.307
≥60	11 (28.2)	152 (36.5)		ref	
**MAC**					
<0.3	12 (30.8)	220 (52.8)	0.009 ^1^	0.40 (0.20–0.81)	0.011
≥0.3	27 (69.2)	197 (47.2)		ref	
**Sex**					
Male	18 (46.2)	195 (46.8)	0.942 ^1^	ref	0.942
Female	21 (53.8)	222 (53.2)		1.02 (0.53–1.98)	
**Age**					
<60	26 (66.7)	242 (58.0)	0.295 ^1^	1.45 (0.72–2.89)	0.297
≥60	13 (33.3)	175 (42.0)		ref	
**Diabetes mellitus**					
Yes	4 (10.3)	60 (14.4)	0.477 ^1^	0.68 (0.23–1.98)	0.480
No	35 (89.7)	357 (85.6)		ref	
**Hypertension**					
Yes	7 (17.9)	105 (25.2)	0.316 ^1^	0.65 (0.28–1.52)	0.319
No	32 (82.1)	312 (74.8)		ref	
**COPD**					
Yes	0 (0.0)	11 (2.6)	0.610 ^2^	N/E	N/E
No	39 (100.0)	406 (97.4)			
**Inhalational anesthetic**					
Desflurane	28 (71.8)	311 (74.6)	0.703 ^1^	ref	0.703
Sevoflurane	11 (28.2)	106 (25.4)		1.15 (0.55–2.40)	
**Operation duration (min)**					
Mean ± SD	83.4 ± 64.6	101.6 ± 81.6	0.271 ^3^	1.00 (0.99–1.00)	0.180
**Anesthesia duration (min)**					
Mean ± SD	121.5 ± 70.7	142.3 ± 90.5	0.319 ^3^	1.00 (0.99–1.00)	0.166
**ASA PS**					
1,2	37 (94.9)	368 (88.2)	0.291 ^2^	2.46 (0.58–10.54)	0.224
≥3	2 (5.1)	49 (11.8)		ref	
**Emergency surgery**					
Yes	2 (5.1)	29 (7.0)	1.000 ^2^	0.72 (0.17–3.15)	0.666
No	37 (94.9)	388 (93.0)		ref	
**Total dose of rocuronium (mg)**					
Mean ± SD	58.3 ± 14.6	61.2 ± 20.5	0.770 ^3^	0.99 (0.97–1.01)	0.400
**Total dose of sugammadex (mg)**					
Mean ± SD	150.0 ± 44.4	152.3 ± 48.3	0.722 ^3^	1.00 (0.99–1.01)	0.777

Values are presented as numbers (percent) or mean ± standard deviation. Abbreviations: SD, standard deviation; T1, time when an inhalation anesthetic was discontinued; T2, time of sugammadex administration; COPD, chronic obstructive lung disease; ASA PS, American Society of Anesthesiology Physical Status; BIS, bispectral index; MAC, minimum alveolar concentration; OR, odds ratio; CI, confidence interval; ref, reference group; N/E, not estimable since none was observed in a certain subgroup. ^1^
*p* values were derived from the Chi-square test. ^2^
*p* values were derived from Fisher’s exact test. ^3^
*p* values were derived from the Mann–Whitney U test. The Shapiro–Wilk test was employed to test normality assumption.

**Table 3 medicina-57-00088-t003:** Clinical factors associated with the T3-T1 (min) in multiple regression analysis (except the duration of anesthesia because of high multicollinearity).

	Uni-Factor Analysis		Multi-Factor Analysis	
Variable	Coef. (95% CI)	*p*	Coef. (95% CI)	*p*
**BIS**				
<60	0.40 (−0.17–0.98)	0.170		
≥60	ref			
**MAC**				
<0.3	−0.14 (−0.69–0.42)	0.623		
≥0.3	ref			
**Sex**				
Male	ref	0.622		
Female	0.14 (−0.42–0.70)			
**Age**				
<60	−0.95 (−1.50 –0.39)	0.001	−0.72 (−1.26–0.18)	0.009
≥60	ref		ref	
**Diabetes mellitus**				
Yes	1.03 (0.23–1.82)	0.011	0.19 (−0.57–0.96)	0.622
No	ref		ref	
**Hypertension**				
Yes	0.19 (−0.45–0.84)	0.553		
No	ref			
**COPD**				
Yes	1.00 (−0.81–2.80)	0.279		
No	ref			
**Asthma**				
Yes	1.27 (−0.53–3.08)	0.165		
No	ref			
**Inhalational anesthetic**				
Desflurane	ref	0.000	ref	0.000
Sevoflurane	2.38 (1.79–2.98)		2.38 (1.80–2.95)	
**Operation duration (min)**	0.01 (0.01–0.01)	0.000	0.01 (0.00–0.01)	0.000
**Anesthesia duration (min)**	0.01 (0.01–0.01)	0.000		
**ASA PS**				
1,2	−0.43 (−1.30–0.45)	0.342		
≥3	ref			
**Emergency surgery**				
Yes	−1.02 (−2.12–0.08)	0.068		
No	ref			
**Total amount of rocuronium (mg)**	0.02 (0.01–0.04)	0.002	0.00 (−0.02–0.01)	0.840

Abbreviations: SD, standard deviation; T1, time when an inhalation anesthetic was discontinued; T2, time of sugammadex administration; T3, time of tracheal extubation; COPD, chronic obstructive lung disease; ASA PS, American Society of Anesthesiology Physical Status; BIS, bispectral index; MAC, minimum alveolar concentration; CI, confidence interval; ref, reference group; Coef., regression coefficient; N/E, not estimable since none was observed in a certain subgroup.

## Data Availability

Please contact the corresponding author for data requests.

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
