# Peer review of "The Relationship between the Timing of Sugammadex Administration and the Upper Airway Obstruction during Awakening from Anesthesia: A Retrospective Study"

_medicina, 2021, doi:10.3390/medicina57020088_

Round 1

Reviewer 1 Report

Please see attached

Author Response

Response to Reviewer 1 Comments

We sincerely appreciate your kind review and reply of our manuscript entitled: “The relationship between the timing of sugammadex administration and the upper airway obstruction during awakening from anesthesia-A retrospective study”.

Your thoughtful suggestions helped us to improve the quality of our manuscript. Our responses to the reviewers’ comment are described below. The manuscript has been revised to reviewers’ suggestions.

We hope the manuscript is accepted for publication in Journal of Medicina.

Sincerely yours,

Major comments 1

Authors have performed a retrospective study entitled ‘ The relationship between the timing of sugammadex administration and the upper airway obstruction during awakening from anesthesia-A retrospective study’ and concluded that incidence of airway obstruction is reduced when sugammadex is administered when MAC of the inhalational agents <0.3 or MAC awake.

Upper airway obstruction can be appreciated only after extubation in general endotracheal anesthesia – hence regardless of time of sugammadex administration, upper airway obstruction will not obvious until extubation. Conclusion of the study can be a bit confusing to readers.

I would rephrase the conclusion – ‘Incidence of upper airway obstruction after successful reversal of muscle paralysis can significantly be reduced when reversal and thus extubation were performed with </= 0.3 MAC’.

Response : Following the suggestion, the manuscript was revised to a clear sentence.

(P9 L237-239)

Major comments 2

Another aspect authors should clarify that even though incidence of upper airway obstruction possibly be less if sugammadex is administered when MAC is </= 0.3, what was the BIS or incidence of recall with that MAC. It might be an ethical challenge to recommend not reversing awake or lightly anesthetized patients.

Response : We fully agree with your opinion for patient safety. Fortunately, there were no patient complained of recall of memory during awakening from anesthesia in this study. Contents about recall of memory have been added to the Results.(P3 L131-132)

Minor comments 1

P1L19 – In abstract, will replace the word general anesthesia with ‘General Endotracheal Anesthesia (GETA)’

Response: The word ‘General anesthesia’ has been replaced to ‘General Endotracheal Anesthesia (GETA)’.

Minor comments 2

P1L36-39 – Redundant since authors are not using anticholinesterase as reversal and TOF is not used in the study

Response: The redundant sentence has been removed.(P1 L37-39)

Minor comments 3

P2L51-52 – Please rephrase the sentence to include extubation as airway obstruction can only happen after extubation, not after sugammadex administration

Response: The sentence has been corrected according to comment.(P2 L48-49)

Minor comments 4

P2L57 – Please use GETA instead of GA only

Response: We changed the word ‘general anesthesia’ to ‘GETA’.(P1 L36, P2 L56, P7 L166, P8 L191,207,217)

Minor comments 5

P2L67 – if TOF was not used, how the dose of sugasmmadex was calculated should be mentioned

Response: Related sentences have been added in Materials and Methods.

The dose of sugammadex was determined by the anesthesiologist taking into time of the last administration of rocuronium, total amount of rocuronium, and the patient’s response.(P2 L70-72)

Minor comments 6

P2L84 – Please omit the word ‘concomitantly’

Response: The word ‘concomitantly’ has been deleted.(P2 L85)

Minor comments 7

P3L106 – Sugammadex was really not prescribed rather administered

Response: The word ‘prescribed’ has been replaced to ‘administered’.(P3 L107)

Minor comments 8

P8L208 – again under discussion, authors should discuss why they think it is okay to wait before reversing muscle paralysis until MAC is 0.3 while extubation can wait unless patient is more awake. Patients’ bucking or coughing may be experienced more in a reversed but still intubated patients but as long as patient is still intubated, airway obstruction is not going to occur. Perhaps, emergence will be less smooth – if that is the reason for late reversal, there should be more detailed clarification and explanation of risks of recall however small it is.

Response: We though that the administration of sugammadex when MAC is <0.3 could recover the patient from neuromuscular block at a time similar to the patient’s recovery of consciousness. But the late administration of sugammadex still has the risk of causing patient to experience about recall of memory, as your thoughtful suggestion.

Even sugammadex was administered when MAC is <0.3, recall of memory was not reported in this study. None of the patients complained of unpleasant memories of tracheal intubation. But it is not a result of questioning all patients. There was a limitation that it is detected only when the patient complained about discomfort.

We respect your suggestion and understand your concerns. However, our study was done with patients in the process of emergence from anesthesia. Even if the patient remembers part of the emergence process, it is different from the arousal during surgery. We carefully suggest that this can be thought of as part of a normal emergence of anesthesia. Of course, it will be difficult to completely rule out the possibility of recall of memory.

In addition, since this study was done retrospectively, there was no intentional delay in administration. If we include the delayed administration of sugammadex in our routine anesthesia process, we will fully explain to the patient the possibility of having an unpleasant memory.

Reviewer 2 Report

While the manuscript is well written and the various measures are well performed, there is an important issue regarding the interpretation of these findings.

Major comment

Why didn’t the authors investigate whether total dose of sugammadex has any correlation with upper airway obstruction after extubation? They should do so.

Minor comment

Legend of table 1, line 1: “Abbreviations: SD, standard deviation; DM” is repeated.

Author Response

Response to Reviewer 2 Comments

We sincerely appreciate your kind review and reply of our manuscript entitled: “The relationship between the timing of sugammadex administration and the upper airway obstruction during awakening from anesthesia-A retrospective study”.

Your thoughtful suggestions helped us to improve the quality of our manuscript. Our responses to the reviewers’ comment are described below. The manuscript has been revised to reviewers’ suggestions.

We hope the manuscript is accepted for publication in Journal of Medicina.

Sincerely yours,

Major comment

Why didn’t the authors investigate whether total dose of sugammadex has any correlation with upper airway obstruction after extubation? They should do so.

Response: Thank you for your comment. We investigated about total dose of administered sugammadex. Total dose of sugammadex did not affect the occurrence of complications. We have added the information about total dose of sugammadex to the table 2 and the results. (P3 L126-127)

Minor comment

Legend of table 1, line 1: “Abbreviations: SD, standard deviation; DM” is repeated

Response: Thank you for your comments. Repeated words have been deleted.(P4 L138)

Round 2

Reviewer 1 Report

Authors have addressed all my comments satisfactorily. In my opinion, it can be accepted for publication.

Reviewer 2 Report

The manuscript has been revised well, the authors are to be commended.